# Evaluation of Tourism Development Efficiency and Spatial Spillover Effect Based on EBM Model: The Case of Hainan Island, China

**DOI:** 10.3390/ijerph19073755

**Published:** 2022-03-22

**Authors:** Pengfei Zhang, Hu Yu, Mingzhe Shen, Wei Guo

**Affiliations:** 1Institute of Geographic Sciences and Natural Resources Research, Chinese Academy of Sciences, Beijing 100101, China; 15530350733@163.com; 2School of Economic and Management, Yanshan University, Qinhuangdao 066004, China; s1269510779@163.com (M.S.); guowei@ysu.edu.cn (W.G.)

**Keywords:** tourism efficiency, spatial spillover effect, sustainable development, Hainan

## Abstract

Tourism development efficiency is one of the key scales to measure the development quality of tourism destination. This study improves the existing input–output index system of tourism efficiency evaluation; knowledge innovation is introduced into the input index, and environmental health pressure is introduced into the output index. Based on the case of Hainan Island, we used the EBM model compatible with radial and non-radial data to evaluate the tourism development efficiency. In order to make up the deficiency of spatial effect analysis based on the geographical distance weight matrix, the spatial spillover effect of tourism development in Hainan Island was analyzed based on a geographical distance weight matrix and an economic distance weight matrix. The findings indicate that nearly 20 years of the Hainan tourism development efficiency mean value was 0.7435, represented by Sanya, and Haikou city of Hainan’s tourism industry development level was higher. However, the spatial spillover effect of Hainan’s overall tourism development is not good. In addition to Tunchang, Ledong city suggests that an appropriate increase in tourism elements, such as investment, expands the scale of the tourism industry, and most cities follow the law of diminishing marginal utility and inappropriate scale blindly. Especially in the face of knowledge innovation becoming the main factor hindering the efficiency of tourism development, we should pay more attention to technological innovation and management reform and coordinate the relationship between tourism development and ecological environment protection.

## 1. Introduction

Amid slow global economic growth and the multiple challenges of COVID-19, the global tourism industry is forming a domestic cycle pattern. Exploring the efficiency of tourism development is to maximize the unit factor input of tourism industry in a specific period of time and maximize the total surplus income of all stakeholders [1]. Tourism development efficiency is a process of continuous evolution from a low level to a high level. The main contradictions and tasks of tourism development efficiency in various countries are different [2]. Compared with developed countries in Europe and the United States, the emerging economies are more dependent on tourism, which requires the coordinated and orderly flow of the tourism industry elements. In order to strengthen the economic attributes of the tourism industry, it is urgent to improve the efficiency of tourism development [1]. Many tourism destinations have experienced the reform process of the tourism development mode, especially in the modern globalization and digital economy society; knowledge productivity gradually plays an important role in the development of the tourism industry, accelerating the reform and renewal of tourism industry development, which is directly related to the development efficiency level of tourism destinations [3]. In addition to paying attention to the development status of tourism in this region, it is necessary to consider the spatial spillover effect of tourism productivity on the tourism productivity of neighboring regions and measure the relationship between tourism performance, growth, and competition at the regional level [4]. The research on tourism efficiency using the DEA method is gradually shifting from industry to geography, and scholars are losing their interest in the accommodation industry, while the research on tourism destinations, especially the sustainability of tourism destinations, is increasingly welcomed among the academic community [5].

In terms of research on tourism development efficiency, the early academic circles mainly focus on the operation and management efficiency of hotels [6,7,8,9,10]. With the continuous development of research, efficiency research has gradually expanded to travel agencies, tourism transportation, scenic spots, and tourism ecology, etc. [11,12,13,14,15,16]. In 1978, Charnes proposed the data envelope analysis method [17], which is widely regarded as a classic method for tourism efficiency research by scholars at home and abroad because it can deal with the problem of multiple inputs and outputs [18]. Especially after Krugman put forward the “Myth of the Asian Miracle” in 1994 [19], efficiency was more widely applied in tourism. In terms of the tourism efficiency measurement, the existing research mostly chooses the tourism fixed asset investment, the number of tourism employees, tourism resource endowment, the number of travel agencies, the number of star hotels, and the number of college students as input indicators [20,21,22] and tourist reception and total income of tourism as output indicators [20,21,22,23,24]. From the macro level of tourism development, the era of simultaneous development of mass tourism and customized tourism has arrived, and the market competition in the tourist destinations is becoming increasingly fierce [25]. The efficiency of the tourism industry proves that there is a complex relationship of interaction, mutual influence, and mutual restriction between input factors and output results [26]. Comprehensive quality of talents, intangible assets, infrastructure, and other factors also play an important role in regional tourism development [27]. From the micro perspective of tourism development, enterprises are an important subject that cannot be ignored in the development of tourism industry, environmental protection, and technological innovation [28], and the difference in innovation modes of tourism enterprises will directly affect the utilization efficiency and energy management ability of tourism resources.

At present, the comprehensive evaluation of regional tourism development efficiency has become a research hotspot in academic circles [29,30,31,32], and based on panel data and through the DEA model, the SBM-DEA model, and the relevant modification model scholars have carried out empirical studies on the industrial development efficiency of tourism destinations [33], the tourism development efficiency of important cities [34], and development, utilization, and protection of core tourism resources [35]. At the same time, the grey correlation analysis and GIS geographic technology [36] are comprehensively applied to explore the influencing factors and spatio-temporal analysis of tourism efficiency in the region. In fact, exploring the problem of development efficiency of tourist destinations is to avoid the decline process predicted by Butler’s life cycle model [27]. To some extent, the efficiency can match the changing situation of the life cycle model of the tourist area and can be divided into a stable model, a reciprocating model, a progressive model, and a radical model according to its efficiency evolution characteristics [34]. Different destinations adopt the expansion of business scale and the improvement in the scientific and technological level to improve the efficiency of tourism development. Destinations with strong resource aggregation capacity and a high utilization level can often achieve higher development efficiency, so as to keep destinations in the upward development stage of their life cycle [35]. At the same time, in the context of continuous scientific and technological progress, the development of different life cycle destinations such as development, stagnation, and recession should be explored at a micro scale [36]. With the emergence of emerging tourism states, scholars began to explore the development efficiency of emerging tourism states, such as cultural tourism [37], ice and snow tourism [38], rural tourism [39], and folk tourism [40]. Throughout the existing literature, there may be three limitations in the study of tourism development efficiency. First, in the selection of research methods for tourism efficiency, most scholars still use DEA models, such as the CCR model and the BCC model [6,7,10,11,12,14,32], while a few scholars use non-radial DEA models, mainly the SBM model [16]. However, both the radial DEA model and the non-radial DEA model have their limitations, the radial model requires the same proportion change of input or output, ignoring the influence of non-radial relaxation variables and failing to achieve the decomposition of efficiency factors [41]. Although the non-radial DEA model could consider all radial and non-radial relaxation variables, it lacks the proportional relationship between the target value of input or output and the actual value [42,43]. Second, in terms of the research content and perspective of tourism efficiency, the current academic circle mainly focuses on the productivity, technical efficiency, and financial efficiency of destinations and tourism operators of different scales from the perspective of economics [44,45]; it advocates changing the extensive growth model driven by tourism elements and realizing intensive and intensional growth. However, such research on tourism performance that ignores the dimensions of resource consumption and environmental loss is flawed [46] and fails to fully grasp the multidimensional connotation of high-quality tourism. Third, in terms of the types of case sites for the research of tourism efficiency, the current academic circle mostly focuses on countries [2,6], regions [11,15,21], provinces [22,23], cities [40], and other regions [12] and pays insufficient attention to the development efficiency of sea-island tourism destinations. In particular, there is a lack of research results to explore the spillover effects of tourism development in sea-island destinations, and the internal mechanism of the development efficiency of sea-island tourism is still unclear.

In general, the existing research on tourism development efficiency pay insufficient attention to sea-island tourism destinations, and the measurement model of tourism efficiency and the selection of unexpected indicators still need to be improved. This study takes Hainan Island as a case study to explore the efficiency and spatial spillover effects of tourism development from 2000 to 2020. The Epsilon-based Measure (EBM) model is used to compensate for the influence of non-radial relaxation variables, and environmental health quality is included in the comprehensive outcome index of tourism development. Taking knowledge innovation as the input evaluation index of tourism development efficiency, the input–output evaluation system was improved, and the spatial spillover effect of each city in Hainan Island was systematically analyzed to clarify the spatial and temporal pattern change of Hainan tourism development efficiency in the past 20 years and analyze the internal factors of its change and development, in order to provide theoretical support for improving the comprehensive benefits of Hainan tourism.

## 2. The Theoretical Analysis

Compared with the industries, the tourism industry belonged to the low energy consumption and green low-carbon industry [47]. Improving the efficiency and quality of tourism development is an effective way to promote the green and high-quality development of China’s economy and society. The theory of the regional economy holds that: in the comprehensive body with vast territory, rich tourism resources, and a complex tourism development mechanism, the development of the tourism economy will produce a significant imbalance due to the differences in resource endowment, location conditions, scientific and technological level, development mode, and other methods [48]. The academic circle has paid attention to the ecological efficiency of tourism development earlier, mainly because the ecological environment and climate change are the prerequisite for sustainable development of tourism [49], and the use of non-renewable resources, resource development and management mode, and ecological footprint and behavior of tourists in the process of tourism development cause many ecological environmental problems [50]. Therefore, scholars have discussed the “relationship between resource utilization, environmental damage and tourism economic benefits” [51].

At present, the academic circle mainly explores the development efficiency of tourism industry at different scales from the perspective of economics. In the process of measuring the efficiency of tourism development, reasonable and scientific input–output indicators are the key steps (Figure 1) [52]. Existing research mainly chose the number of tertiary industry employees, tourism resource endowment, the number of star hotels and travel agencies, and the investment amount of urban fixed assets as the input indexes to evaluate the efficiency of tourism development and selected the total income of tourism and the number of tourist reception as the output indexes. The influencing factors mainly explore the effects of tourism resource endowment, industrial structure, economic development, and traffic location conditions [23]. Tourism is not only intertwined with regional economy but also cooperated with each other to produce network relations. At the same time, the development of tourism needs to consume the natural capital of tourist destinations, and the negative effects of carbon emissions, waste water, waste gas, waste solids, and other effects generated by tourism activities affect the ecological environment of tourist destinations [53]. The development of tourism relies on favorable ecological environment system conditions, which also forces tourism destinations to implement relatively strict environmental regulation policies [54].

Regional economic growth is influenced by the development level of tourism in the region, the development level of tourism in neighboring regions, and the economic status [55]. The spatial spillover effect of tourism development is mainly manifested in two aspects: the first is the spatial spillover effect of the tourism economy itself, mainly manifested in the way of tourism flow, whose essence is the regional liquidity of capital [48]. Regional tourism flow has the possibility of spreading neighboring regions, and potential tourism flow has a positive effect on neighboring tourism attraction and tourism development, that is, the development of regional tourism will promote the development of neighboring tourism [56]. The upward and downstream extension of the tourism industry chain further provides the possibility of industrial connection for the spatial spillover of regional tourism development [48]. Secondly, the tourism industry is highly dependent. Relying on the regional sharing mechanism of tourism development elements [57], tourism development is inseparable from infrastructure and human capital. Capital, human capital, technology, and other elements may share development in neighboring regions. Existing studies have confirmed that infrastructure, human capital, and other factors have significant spatial spillover effects on tourism economic growth. At the same time, the ecological environment of the tourism economy also has a spatial spillover effect, and the regional industrial structure produced by tourism economy will have an impact on the ecological environment. The joint prevention and control measures adopted in the joint development of regional tourism will also control the ecological environment. The relationship between ecological and environmental effects of tourism development in neighboring regions is also dialectical. For example, empirical studies have confirmed that per capita tourism consumption and total tourist trips in China effectively promote carbon emissions in their own provinces but have an inhibitory effect on carbon emissions in neighboring provinces [58].

## 3. Methodology and Data Sources

### 3.1. Evaluation of Urban Tourism Development Efficiency

#### 3.1.1. Index System Construction

Based on the existing research results [39,41,46] and the availability and operability of data, objective and scientific indicators are selected to constitute the evaluation index system of tourism development efficiency of Hainan Island. Based on the relationship between various inputs and expected and unexpected outputs involved in the Epsilon-based Measure (EBM) model, some alternative indicators are selected to improve the input–output index system of total factor productivity for tourism destination development (Table 1) [59]. Labor, capital, land, and other factors are the basic indicators to evaluate the efficiency of tourism development. However, due to the difficulty in obtaining the data of tourism land use area, land is hardly used as an input indicator to evaluate the efficiency of tourism development in the existing theoretical studies. This study selected the number of tertiary industry employees, investment in urban fixed assets, the number of star-rated hotel rooms, tourism resource endowment, and knowledge innovation as the input indexes to evaluate the tourism development efficiency of Hainan Island. Among them, knowledge innovation is replaced by the number of high school students per 10,000, indicating the talent reserve and knowledge progress of tourism development. The tourism resource endowment is judged by the scale and grade of a-class scenic spots. As tourism is a comprehensive industry, the number of employees in the tertiary industry, the input of urban fixed assets, and the input of knowledge innovation are partially deviated from the actual scale of the input factors of tourism, but they also include the input factors of tourism development to a large extent [59,60]. Tourist reception and total income of tourism were selected as output indexes of tourism development efficiency to measure the scale and benefit level of Hainan tourism development [22]. At the same time, considering the ecological environment pressure brought by tourism to the destination, environmental health quality is regarded as the unexpected output of tourism development [46], which is also in line with the social trend of low-carbon and green development and can better measure the comprehensive benefits brought by tourism development to the destination. Its environmental health quality assessment methods refer to the research methods of Scholars such as Schandl H [61] and Xiao Z [62] in the relevant literature.

#### 3.1.2. Measurement of Tourism Efficiency

Taking Hainan Island as the case study, this study introduces environmental health quality factors based on the traditional concept of “tourism development efficiency” and redefines it as “Full efficiency of Tourism Development (FEOTD)”. This was done while pursuing the sustainable growth of the tourism economy, to minimize the environmental health quality pressure caused by waste water and exhaust gas emissions from tourism development to destinations [41]. Suppose the tourism output value of each city in Hainan Island is Y; the input vector of tourism industry development factors is X = (X_1_, X_2_,…, X_m_); and m is the number of input indicators. Referring to Kuosmanen’s research results [63], the full efficiency of tourism development can be expressed as follows:(1)$$\mathrm{FEOTD}=y/\sum_{i=1}^{m}\omega_{i}x_{i}$$
where FEOTD is the full efficiency of tourism development, $\omega_{i}$ (*i* = 1, 2,…, M) is the input weight of each tourism factor, satisfying $\sum \omega_{i}=1$. Based on the research results of Kuosmanen [63], Picazo-Tadeo [64], and other scholars, this study introduces the EBM model to evaluate the tourism development efficiency of Hainan Island and gives full play to the advantages of this model in taking into account both radial and non-radial relaxation variables. Cities to be evaluated in Hainan Island ο (ο = 1, 2,…, 17). The linear normative expression for the measurement of tourism development efficiency is as follows [41]:(2)$$\min\gamma^{*}=\theta -\varepsilon_{x}\sum_{i=1}^{m}\frac{\omega_{i}\bar{s_{i}}}{x_{io}}$$
(3)$$s.t.X\lambda -\theta x_{io}+\bar{s}=0,Y\lambda \geq y_{o},\lambda \geq 0,\bar{s}\geq 0$$

Type in the *X*, *Y*, λ, and $\bar{s}$ for the input, output, weight coefficient, and input slack variables, respectively; $\gamma^{*}$ is a review of the city of tourism development of the whole efficiency of value; θ is the radial composition of $\gamma^{*}$; $\varepsilon_{x}$ is the key parameter of value between 0 and 1, said the radial part in the important degree of the efficiency value, the value of 0 and 1; the model is equivalent to the radial model and the SBM model, respectively. Refer to Tone [43] for specific operation process.

If “∑λ=1” is added to Formula (2), the market to be evaluated can be evaluated ο (ο = 1, 2,…, 17). The “pure technical efficiency” of tourism industry development, the ratio of the total efficiency, and the pure technical efficiency of tourism industry development is the “scale efficiency” of tourism industry development in the city. In this study, the EBM model based on input-oriented and constant return to scale is abbreviated as EBM-I-C, and the EBM model based on input-oriented and variable return to scale is abbreviated as EBM-I-V [41].

### 3.2. Spatial Panel Econometric Model

#### 3.2.1. Space Weight Matrix Settings

(1) Spatial weight matrix of geographical distance. According to the “first law of geography,” there is a relationship between something and its surroundings within a certain area, and the relationship becomes stronger the closer you are. In order to describe the distance attenuation characteristics of spatial distance influence, the weight is set according to the reciprocal geographical distance between two cities, and the spatial weight matrix of geographical distance is constructed, and the formula is as follows [2]:(4)$$W_{ij}=\left\{ \begin{array}{l} 1/S_{ij},i\neq j\\ 0,i=j \end{array}\right.$$

(2) Weight matrix of economic distance space. There may be deviation in measuring weight solely by geographical space distance. Therefore, considering the influence of factors such as economy, culture, and system, we can refer to the spatial weight matrix of economic distance nested between regional economic differences and geographical space weight matrix proposed by Huogen W [65]. It is assumed that the urban complex with strong economic strength exerts stronger force on neighboring cities, while the urban complex with weak economic strength exerts weaker force on neighboring cities, and the formula is as follows [2]:(5)$$W_{ij}=W_{1}\times diag\left(\frac{\bar{Y_{1}}}{\bar{Y}},\frac{\bar{Y_{2}}}{\bar{Y}},\cdots ,\frac{\bar{Y_{n}}}{\bar{Y}}\right)$$
(6)$$\bar{Y_{i}}=\frac{1}{t_{1}-t_{0}+1}\sum_{t=t_{0}}^{t_{1}}Y_{it},\overset{=}{Y}=\frac{1}{n(t_{1}-t_{0}+1)}\sum_{i=t_{0}}^{t_{1}}\sum_{i=1}^{n}Y_{it}$$
where $W_{ij}$ is the spatial weight between i city and j city; $W_{1}$ is the geographical space weight matrix; $diag\left(\frac{\bar{Y_{1}}}{\bar{Y}},\frac{\bar{Y_{2}}}{\bar{Y}},\cdots ,\frac{\bar{Y_{n}}}{\bar{Y}}\right)$ is the diagonal matrix in which the mean proportion of GDP of each city is the diagonal element; $\bar{Y_{i}}$ represents the mean value of real GDP of i city in each year; $\overset{=}{Y}$ represents the real GDP of all cities; $t_{0}$ is the base period of the study; $t_{1}$ is the end of the study; and $Y_{it}$ is the real GDP of i city in t year.

#### 3.2.2. Selection Method of Space Panel Metering Model

Firstly, Moran’s I, LHoisting spatial autocorrelation test was selected, and LM-ERROR and LM-lag were used to judge the existence of error terms or lag terms in spatial correlation. If both conditions exist, then LR-LAG and LR-error were used to judge whether the space panel Dubin model (SPDM) was selected and, combined with Robust LM lag and Robust LM error, whether SPDM could be simplified into the space panel error model (SPEM) and the space panel lag model (SPLM) [66].

#### 3.2.3. Spatial Effect Decomposition

Point estimation may lead to parameter estimation error, which can be compensated by the partial differential method, which can decompose the estimated results into direct effect, indirect effect, and total effect. Taking SPDM as an example, it can be transformed into $(I_{n}-\rho W)=\iota_{n}\mu_{0}^{\prime }+\beta X+\theta WX+\varepsilon$, order, and $Q_{m}(W)=P(W)\times (I_{n}\beta_{m}+\theta_{m}W)$; it can be transformed into:(7)$$Y=\sum_{m=1}^{k}Q_{m}(W)X_{m}+P{\left(W\right)}_{\ln}\beta_{0}^{\prime }+P(W)\varepsilon$$

Then, convert it to the matrix form as follows:(8)$$\left[\begin{array}{l} Y_{1}\\ Y_{2}\\ \cdot \\ \cdot \\ Y_{n} \end{array}\right]=\sum_{m=1}^{k}\left[\begin{array}{cccc} Q_{m}{\left(W\right)}_{11} & Q_{m}{\left(W\right)}_{12} & \cdots & Q_{m}{\left(W\right)}_{1n}\\ Q_{m}{\left(W\right)}_{21} & Q_{m}{\left(W\right)}_{22} & \cdots & Q_{m}{\left(W\right)}_{2n}\\ \cdot & \cdot & \cdot & \cdot \\ Q_{m}{\left(W\right)}_{(n-1)1} & Q_{m}{\left(W\right)}_{(n-2)2} & \cdots & Q_{m}{\left(W\right)}_{(n-1)n}\\ Q_{m}{\left(W\right)}_{n1} & Q_{m}{\left(W\right)}_{n2} & \cdots & Q_{m}{\left(W\right)}_{nn} \end{array}\right]\left[\begin{array}{c} X_{1m}\\ X_{2m}\\ \cdot \\ \begin{array}{l} X_{(n-1)m}\\ X_{nm} \end{array} \end{array}\right]+P(W)(\tau_{n}\beta_{0}^{\prime }+\varepsilon )$$
where *m* represents the M_th_ explanatory variable, m = 1, 2,…, k. The first matrix on the right of the equal sign is the partial differential matrix. The element on the diagonal represents the average influence of $X_{ik}$ city’s variable change on the explained variable of neighboring cities, that is, the indirect effect is the spatial spillover effect, $indirect=\partial Y_{i}/\partial X_{jm}=Q_{m}{\left(W\right)}_{ij}$. The total effect is the synthesis of direct effect and indirect effect, $total=Q_{m}{\left(W\right)}_{n}+Q_{m}{\left(W\right)}_{ij}$ [2].

### 3.3. Data Source and Processing

In this study, the accessibility and authority of data were fully considered. Data of cities in Hainan Island from 2000 to 2020 were mainly derived from relevant yearbook data collected by the China Economic and Social Big Data Research Platform, such as the Hainan Statistical Yearbook, the Haikou Statistical Yearbook, and the China Urban Statistical Yearbook. Part of the data comes from the official websites of the Hainan Municipal Statistical Bulletin on national economic and social development and government statistics bureau. In order to ensure the integrity of the research data, the SPSS linear interpolation method was used to supplement the missing values. Due to the serious lack of relevant data of Sansha City and Baisha Li Autonomous County, this case was not considered in this study.

## 4. Results Analysis

### 4.1. Evaluation Results of Hainan Tourism Development Efficiency

#### 4.1.1. The Changing Trend and Regional Difference of Tourism Development Efficiency

In order to avoid the problem of low regional efficiency difference caused by fewer decision-making units and sparse data, this study adopted the window analysis method to construct the production front and took the input–output data of the first S period including the current period as the reference technology set, so as to reduce the calculation period by S-1 cycle. The window width set in this study was 2, that is, the reference technology of each year was jointly determined by the input–output values of the current period and the early period. According to Formula (2), the EBM-I-C model results of tourism development efficiency of 17 cities in Hainan Island can be calculated (Table 2). Meanwhile, to facilitate comparison of model results, this article presents the results of the tourism development effect of Hainan Island measured by the CCR-I-C model and the SBM-I-C model [41].

The results in Table 2 show that the efficiency value measured by the EBM-I-C model was 0.7435, which was just between that measured by the CCR model and the SBM model. Considering the accuracy and scientificity of the EBM model, this study used the EBM model to calculate and numerically analyze the characteristics of spatial and temporal differences of tourism development efficiency in Hainan Island. From 2001 to 2020, the average efficiency of tourism development in Hainan Island was 0.7435, indicating a high level of tourism development, but there is still a large space for improvement and optimization. In other words, the tourism development elements in Hainan Island have not reached the optimal matching operation state, and some elements are lagging behind. According to the results of efficiency decomposition, the mean values of pure technical efficiency (PTE) and scale efficiency (SE) of tourism development in Hainan Island are 0.9406 and 0.7881, respectively. There is still much room for improvement in scale efficiency (SE), while the mean value of pure technical efficiency (PTE) is high, close to the optimal efficiency value 1, and significantly better than the mean value of scale efficiency (SE). Therefore, in order to improve the efficiency of tourism development in Hainan Island, on the premise of ensuring the basic pace and path of tourism development, we should restore and expand the scale of tourism development in Hainan Island in an orderly manner according to the actual conditions of regional tourism development in Hainan Island and give full play to the scale effect of tourism development in Hainan Island.

From the perspective of the time series development trend, Hainan’s tourism development efficiency fluctuated from 0.6561 in 2001 to 0.7010 in 2020. The decomposition of pure technical efficiency and scale efficiency in development efficiency shows that the improvement in the tourism development efficiency in Hainan Island depends more on scale efficiency. It should be noted that the tourism development efficiency of Hainan province in 2020 is lower than that of 2019, mainly due to the impact of COVID-19. According to the Statistics Bureau of Hainan Province of China, the number of visitors and the total income of Hainan tourism decreased by 22.3% and 17.5%, respectively, in 2020. Only from January to April 2020, the total number of tourists received in Hainan province decreased by 58.4% during the same period, which is clearly demonstrated in the measurement of tourism efficiency. Therefore, all cities in Hainan Island should pay more attention to the rational use of tourism industry elements, fully improve the level of tourism management and technology application, continue to develop the scale efficiency of tourism, control the environmental impact of tourism pollutant emissions on tourism destinations, and promote the intensive use and green high-quality development of Hainan tourism destinations.

The decomposition results of tourism development efficiency in Hainan Island from 2001 to 2020 are shown in Table 3. Among the 17 urban areas studied in Hainan Island, Sanya always takes the lead, and its tourism development efficiency value is close to 1. The tourism development of Haikou, Wenchang, and Wanning and efficiency value in 2–4 places to settle; the Tunchang county and Ledong county tourism development level was low efficiency, with a serious lag behind the overall average efficiency of Hainan tourism development level; the tourism development efficiency values were below 0.6, and the tourism development of technical factors and the scale effect was limited. The driving effect of tourism on local economic and social development is not significant. According to the efficiency decomposition results, Sanya is the city with the highest pure technical efficiency in Hainan Island, with a long history of tourism development, mature and perfect supporting facilities and application technologies for tourism development and strong agglomeration capacity of tourism industry [67]. It is a typical region for tourism development in Hainan Island and even the whole country. Qiongzhong county and Baoting County are also cities with high pure technical efficiency in Hainan Island. Although the tourism industry in Qiongzhong and Baoting developed late, their tourism resources are deep. With the continuous improvement in the tourism promotion plan, especially the transportation network, the technological progress and management of the tourism industry in Qiongzhong and Baoting have been effectively improved. Statistics show that Baoting county and Qiongzhong County have the lowest environmental quality pressure brought by tourism development, that is, the total amount of pollutants such as sewage discharge, solid waste, and exhaust emission brought by tourism is relatively low. In terms of the environmental pollution emission effect of the unit tourism output value, Lingao county and Tunchang County pay less environmental pollution cost of unit tourism output value.

In contrast, Danzhou city, Qionghai City, and Dingan County have the lowest pure technical efficiency among the 17 cities in Hainan Island, with the pure technical efficiency value below 0.8. Danzhou’s tourism industry is monotonous; technical progress and the management level is limited; there is the tourism industry transformation and the upgrading of the development of technical management problems. The tourism development foundation of Qionghai city and Dingan County is relatively weak, and the tourism development model needs to be further optimized. Especially, the tourism development of Qionghai City brings great pressure on environmental quality, and the original applied technology and efficiency are difficult to meet the new requirements of tourism development. All these hinder the improvement in the pure technical efficiency of urban tourism development. The cities with the highest scale efficiency are Haikou and Sanya. The tourism input factors are large, and the tourism industry is mature. The scale efficiency and technical efficiency are close to 1, indicating a high matching degree. The scale efficiency of the Wenchang, Wanning, and Chengmai tourism industry development is also at a high level, with the scale efficiency value above 0.9. The scale of the tourism industry and the environmental pollution pressure are basically reasonable.

It should be noted that among the 17 cities in Hainan Island, Tunchang County has the lowest scale efficiency value of tourism development, at only 0.5832. This is mainly because Tunchang County’s tourism industry development is relatively backward, with a lack of core tourism attractions, and the tourism industry development facilities are not perfect, greatly restricting the expansion of the development of the tourism industry. According to the statistics, Haikou and Sanya occupy a leading position among all the cities in Hainan island, both in terms of pure technical efficiency and scale efficiency, and they have relatively advanced industrial development models and concepts. Due to the law of diminishing marginal returns, too much resource investment will lead to the decline in economies of scale, thus restricting the improvement in the development efficiency of the tourism industry [41]. At the same time, Wuzhishan city, Dingan County, Tunchang County, Changjiang county, and Ledong County’s tourism industry development scale efficiency value is also low, which is the main factor affecting the efficiency of tourism development in these areas. However, tourism development in these cities is still in the stage of increasing returns to scale, so it is necessary to properly increase the input of tourism development factors and improve the application level of tourism industry technology, so as to effectively bring into play the economies of scale of the tourism industry.

From the perspective of temporal and spatial changes in tourism development efficiency in Hainan Island from 2001 to 2020, nodes with an interval of about 5 years were selected for analysis to ensure a significant comparison effect (Figure 2). From the perspective of the overall regional effect, the tourism development efficiency of Hainan Island has been steadily rising. Except Sanya and Haikou, the tourism development efficiency of the north and south poles has been leading the overall development, and the pattern of tourism development efficiency has been high in the northeast and low in the southwest. It should be noted that the overall decrease in Hainan’s tourism development efficiency in 2020 is mainly caused by COVID-19, rather than a dilemma in Hainan’s tourism development. This has been verified in theoretical research and the practical development of many tourist destinations around the world. The empirical results show that although the tourism development efficiency of southwest China represented by Changjiang and Tunchang has been significantly improved, the overall tourism development effect of Hainan Island has performed well, especially after 2015. Due to the late start of tourism, the insufficient exploitation of tourism resource endowment, and the small brand influence, there is always a development gap with the northeast region of Hainan Island.

Hainan Island opened the whole railway around the island in 2015, but according to the evolution chart of tourism development efficiency, the tourism development efficiency of Hainan Island cities has been improved, but there is still more room for improvement, and the tourism development network around the island needs to be strengthened. In addition to the need to strengthen the construction of island tourism, we should also carry out targeted strengthening and improving policies according to the efficiency of tourism development to strengthen the cooperation between Hainan Island’s northeast region represented by Haikou and Wenchang and southwest region represented by Dongfang and Ledong, to strengthen the misallocation of tourism resources, and to create high-quality tourism portfolio products. Although currently facing the COVID-19 outbreak of uncertainty, Hainan Island should always pay attention to the tourism potential after mining and epidemic development work, especially on the policy of the free trade port in Hainan opportunities; regard Hawaii, Bali, and other world-famous tourism destinations as the development target; and build “Hainan international tourism island” brand strength and international status.

#### 4.1.2. Potential Index of Tourism Development Efficiency Improvement in Hainan Island

This study uses the EBM model to measure the gap between the target value of each input and the actual value to measure the tourism development efficiency improvement potential index of Hainan cities [41], and the calculation results are shown in Figure 3. The result shows: the municipal tourism development of Hainan Island inputs as potential space of different sizes, from inputs as potential space size sorting, Knowledge innovation > Tourism resources endowment of fixed assets investment > Investment in urban fixed assets > Number of star hotel rooms > Number of tertiary industry employees; the knowledge innovation is the chief factor that restricting tourism development of Hainan Island efficiency overall. Due to the long history of tourism development in Hainan Island, the tourism development model and path are mature and stable, but the effect of technological input factors and the management progress is not very significant, and there are still deficiencies in the construction of high-level tourism industry management personnel and the introduction of advanced technology in the field. Tourism development factors such as tourism resource endowment and investment in urban fixed assets are characterized by strong stability, and the significant change in tourism development efficiency is seldom realized due to the sharp increase in a certain factor. The tourism industry application technology, management personnel construction, and other aspects have the characteristics of soft power, so it is easier to highlight its role in the overall process of tourism industry development.

At the regional level, Haikou city, Sanya City, and Wenchang City have the lowest spatial index of tourism development promotion potential, with the average spatial index of each factor being 4.51, 1.56, and 1.16, respectively. Haikou and Sanya have developed tourism and are the pillar cities of Hainan tourism. It is obvious that the potential of tourism development and promotion is small. While Wenchang situation has the obvious difference with the former two, Wenchang is located in the eastern coastal areas of Hainan Island, with developed traffic network; its tourism brand is inferior to Haikou and Sanya, but by focusing on creating space tourism, coastal tourism, ecology, tourism, and other special products, it is inside the area to better match the elements of a relationship, and the overall level of tourism development efficiency is higher. Qionghai city, Danzhou City, and Dingan County have the largest potential for tourism development. The factors that affect the efficiency of tourism development in these three cities are different. Tourism resource endowment, investment in urban fixed assets, and knowledge innovation are the primary factors that affect the efficiency of the tourism industry in these cities. It is an important way to strengthen the tourism development efficiency of these cities to strengthen the investment of tourism capital, improve the application technology and management level of tourism industry, and continuously refine and enhance the radiation power of tourism core attractions.

### 4.2. Spatial Spillover Effect of Tourism Development Efficiency

#### 4.2.1. Model Setting and Validation

First of all, the development efficiency of the tourism industry was the explanatory variable; the total tourism revenue was the core explanatory variable; and the urban fixed asset investment, the tertiary industry staff number, the star hotel rooms, the tourism resource endowment, and knowledge innovation were taken as the control variables. It should be noted that the innovation development potential and the level of the tourism industry measured from the perspective of innovation input [68] is also a direct factor to promote the transformation and development of the tourism industry. According to the test of the variance inflation factor (VIR), the maximum value of each variable is less than 5, which is suitable for constructing spatial panel measurement model. At the same time, the unit root test was conducted, and the results showed that all variables LLC and Fisher-ADF test rejected the null hypothesis at the level of 0.05, indicating that all variables were stable and that parameter estimation could be performed directly [2]. The explained variable tourism development efficiency was replaced by symbolization as lnTDE. Moran’s I test reached the significance level of 0.05; the Lrations and Walds test reached the significance level of 0.01; the LM-ERROR test was significant at the level of 0.1; and the LM-LAG test was significant at the level of 0.01. The results of LR-LAG and LR-error tests showed that the results were significant at the level of 0.05 and 0.01, respectively, that is, the SPDM model should be selected finally.

#### 4.2.2. Estimation of Tourism Development Effects

MATLAB was used to estimate parameters under two kinds of spatial weights. First, the Hausman test significantly rejected the null hypothesis, indicating that fixed effect estimation should be used. To avoid the influence of unobserved urban heterogeneity factors and time changes on the estimation results, the spatio-temporal double fixed estimation was set [69]. To make it easier to measure, in this study, investment in urban fixed assets, the number of tertiary industry employees, the number of star hotel rooms, the tourism resources endowment, and the knowledge innovation were symbolized as lnIIUFA, lnNOTIE, lnNOSHR, lnTRE, and lnKI, respectively. The results are shown in Table 4: In terms of the impact of tourism industry input factors on tourism development efficiency, the lnIIUFA coefficient was 0.5098 under the geographical distance weight matrix, and the significance level of 0.05 indicates that urban fixed asset input factors can effectively improve the efficiency of the tourism industry. The coefficient of the spatial lag term W*lnIIUFA was −2.6000 and failed to pass the significance level, indicating that the spatial spillover effect of urban fixed asset input factors on adjacent areas was not significant, and its effect on improving the development efficiency of tourism industry in adjacent areas was limited. The estimation results of economic distance weight matrix were similar, which confirms the robustness and validity of the above results. It should be noted that Lesage believes that there may be some errors in the evaluation of the spatial spillover effect by point estimation [70], so the above results are preliminary estimates of the spatial spillover effect of tourism development efficiency.

Under the two types of spatial weight matrices, the total effect of tourism industry input factors on tourism industry development efficiency was decomposed into the direct effect and the indirect effect, and the results of micro-partial estimation were analyzed, as shown in Table 5.

(1) The impact of tourism industry input factors on local tourism development efficiency. Under geographical distance weight matrix, lnIIUFA, lnTRE and lnKI coefficients were 0.8603, 0.8425 and 1.4576, respectively, and passed significance level tests of 0.01, 0.01 and 0.05, respectively. It shows that Investment in urban fixed assets, Tourism resource endowment and Knowledge innovation play a more significant role in improving the efficiency of local tourism industry. This may be due to the following two points: one is the Tourism resources endowment is the basis of regional tourism, is the core elements within the system of tourist attractions, and Investment in urban fixed assets is necessary to improve the tourism facilities to provide important support, the lack of perfect supporting facilities will not be able to carry out a number of tourism activities. Knowledge innovation is the key means of tourism industry development management, with advanced management skills and ideas to constantly adjust and optimize regional tourism development efficiency; Secondly, although the number of tourism employees and the number of star-rated hotel rooms are also important investment indicators, due to the seasonal characteristics of tourism, the scale of tourism employees is not very stable, and can not be effectively measured as a robust factor. The number of star-rated hotel rooms is accompanied by the continuous enrichment of accommodation forms, and the attraction and reception capacity of star-rated hotels have decreased, so its effect on the development efficiency of the local tourism industry is limited. Under the weight matrix of economic distance, only the lnTRE coefficient is 0.9201 and passes the 0.01 significance level test, indicating the prominent role of tourism resource endowment, which also verifies the value judgment that tourism resource endowment is the basis of regional tourism development.

(2) Analysis of spatial spillover effect of tourism industry input factors on adjacent areas. Under the geographical distance weight matrix, the lnNOTIE and lnTRE coefficients were positive, 3.3518 and 1.1754 respectively, and all passed the 0.01 significance level test. The results show that the number of tertiary industry employees and tourism resource endowment in Hainan Island have a positive spatial spillover effect on the improvement of tourism industry development efficiency in neighboring regions, and tourism employees have high regional and industrial mobility, which is extremely beneficial to supplement tourism employees in neighboring regions. The optimal combination of local and adjacent tourism resource endowment can further strengthen the attraction of tourism industry in the region, which is very beneficial to the development of regional tourism.lnIIUFA and lnKI coefficients were negative, −1.4939 and −2.5965 respectively, and passed the significance level test of 0.05 and 0.1. This indicates that the urban fixed asset input and knowledge output of Hainan Island have not yet formed a good spatial spillover effect, and even have a restraining effect on neighboring regions to a certain extent. This is not only related to the incomplete connection network between cities and the unsmooth flow of tourism industry input factors, but also related to the tourism development competition of neighboring cities. Since Hainan’s tourism industry is mainly coastal tourism, vacation tourism and ecological health tourism, and some tourism resources are seriously homogenized, the inter-regional tourism development also has fierce competition, which will undoubtedly have a negative effect on the spatial spillover effect of neighboring regions. Economic distance weighting matrix, the urban fixed asset investment, tourism resources endowment and the adjacent areas produce positive spillover effect on knowledge innovation, star hotel guest room number of neighboring regions produce negative spillover effect, and this geographic distance weighting matrix under there was a discrepancy in the performance results, mainly due to geographical distances and economic distance is different as a result of the emphasis, The same is that tourism resource endowment has a significant spatial spillover effect on the tourism development of neighboring regions.

## 5. Discussion

Tourism development efficiency is an important theme concerned by tourism destinations around the world [20,25,28,35]. With the different development stages of tourism destination’s life cycle, the development efficiency of tourism industry will have different manifestations, such as sustained development, stagnant development or recession. Under the promotion of the policy of building an international tourism island and a free trade port issued by the State Council of China, building an “international tourism and consumption center” has become the strategic goal of Hainan Island’s tourism development. Hainan Island is China’s tourism industry to carry out early and become one of the typical development of tourist destination, is a more traditional tourist destination in China, southeast Asia area, this with Paris [71], London [72], Singapore and other world famous tourist destination development efficiency has the same place, the development of the famous tourist destinations are facing serious challenges, That is to break through the traditional tourism development path so as to further improve the development efficiency of tourism industry and give full play to the comprehensive driving role of tourism in the economy and society. In this paper, investment in urban fixed assets, number of nertiary industry employees, number of star hotel rooms, tourism resources endowment and knowledge innovation are selected the input indexes of tourism development efficiency. Taking tourist reception, total income from tourism and environmental health quality as output indexes, the efficiency level and spatial spillover effect of tourism development in 17 cities in Hainan Island were analyzed. In order to provide reference for the research and construction of other tourist destinations in the world.

After considering the comprehensive benefits of tourism development, tourism environmental health quality results will be included into the evaluation standard system, emphasizes the development of tourism can not simply focus on economic benefit while ignoring the ecological environment quality, adhere to the good ecological environment is the basic condition of developing tourism in view and theory of [73]. At the same time, this paper takes knowledge innovation as the input factor of tourism industry development efficiency evaluation, pays attention to tourism application technology, management mode and talent training, and emphasizes the modernization role of tourism development platform and tools, which is the improvement of the input-output index system of tourism development efficiency. After constructing a reasonable evaluation index system, EBM model was selected in this paper, including radial and non-radial distance functions, which made up for the deficiencies of traditional DEA model and non-radial SBM model to some extent [74]. The traditional radial DEA model requires that all inputs and outputs change in the same proportion and ignores the influence of non-radial relaxation variables. The non-radial model takes into account all radial and non-radial relaxation variables, but loses the proportion information between the input or output target value and the actual value. EBM model can solve this problem effectively. This made the evaluation results of tourism development efficiency in Hainan Island more objective and enriched the application of EBM model in tourism. In the analysis of spatial spillover of tourism development efficiency in Hainan Island, this paper chooses the weight matrix of geographical distance and the weight matrix of economic distance for joint analysis, so as to avoid the result deviation caused by ignoring economic and social effects caused by relying solely on geographical distance, and enrich the measuring scale tool of spatial spillover effect.

By measuring the tourism development efficiency and spatial spillover effect of Hainan Island in the past 20 years, the paper comprehensively shows the development trajectory of Hainan Island as a famous traditional tourism destination in both time and space dimensions, and objectively analyzes the main factors restricting the improvement of tourism efficiency of Hainan Island from the input index. This provides theoretical reference for the improvement of development efficiency and path breakthrough of world-famous tourism destinations, especially the identification of turning points of development efficiency of tourism industry in different cities on the time trajectory, and the strengthening of interaction between neighboring tourism destinations on the spatial scale. From the empirical results of Hainan tourism industry development efficiency, pure technical efficiency of 17 cities tourism development of Hainan Island is not very significant difference, Haikou, Sanya as the advanced on behalf of the size differences in different areas of outstanding, Wuzhishan, Dingan, Tunchang county’ tourism industry scale is still been in stage, Appropriate increase of tourism industry input factors and expansion of tourism industry scale are the main means to improve the efficiency of tourism development in these cities. From the perspective of spatial spillover effect of tourism industry development efficiency, the positive promotion effect of tourism development in 17 cities in Hainan Island is limited, and some development factors among neighboring cities even play a negative role. Strengthening the regional cooperation mechanism with tourism resource endowment as the core, paying attention to the ecological environment construction in Hainan, and matching it with appropriate macro-adjustment policies [75] are the best way to continuously promote the harmonious coexistence between man and land in tourism destinations and achieve sustainable development.

In the face of the disruptive impact of COVID-19, the impact degree and performance of cities in different life cycle stages vary, which may range from short-term tourist loss to permanent departure of tourism [76]. However, in summary, all of them will exert a restraining effect on the development of tourism destinations. Hainan Island, as a destination with tourism as an important economic support, is more sensitive, and tourists’ demands cannot be effectively met, resulting in the life cycle of its tourism destination moving in the direction of recession [77]. However, in terms of actual development, under the condition that the COVID-19 epidemic is basically under control, Hainan’s tourism development shows the characteristics of rebound growth, that is, its tourism recession is an illusion of development caused by unavoidable external factors. In response to the impact of COVID-19 on tourism, China and Hainan province have introduced tax, financial support, rent reduction and other measures to provide capital for the development of Hainan’s tourism industry in the post-epidemic era, so as to prevent the development of Hainan’s tourism industry from falling into a state of complete recession that cannot be recovered. In the future, the tourism development of Hainan Island should focus on the role of scientific and technological innovation, pay more attention to the construction of domestic tourism market in the post-epidemic era, and strive to build an “international tourism consumption center”. Under the premise of controllable epidemic, the international tourism market in Europe and North America should be reasonably expanded to improve the competitiveness and attractiveness of Hainan’s tourism brands in the international market.

## 6. Conclusions

### 6.1. Concluding Remarks

The evaluation of tourism development efficiency should avoid single evaluation based on the data of a certain year, but should carry out long-term tracking and exploration of tourism destination. Hainan Island is one of the classic tourist destinations in China and even in Southeast Asia. Using EBM model to explore the change of tourism development efficiency in Hainan Island in the past 20 years has strong theoretical and practical value. From 2001 to 2020, the average tourism development efficiency of Hainan Island is 0.7435, indicating a high level of tourism development, but there are many problems of tourism efficiency optimization at the same time. From the perspective of contributors to the improvement of tourism development efficiency in Hainan Island, the pure technical efficiency of tourism development in Hainan Island basically remains stable. The scale efficiency fluctuates from 0.6697 in 2001 to 0.8392 in 2020, which is the main reason for the improvement of tourism development efficiency in Hainan Island. At present, it is difficult for Hainan Island to ensure the sustainable development of tourism solely relying on the resource concept of “tropical islands”, and the tourism development factors have not reached the optimal matching structure relationship, and some of the tourism development factors even lag behind the level of system development equality. From the perspective of urban tourism development level, Sanya and Haikou, with their rich tourism resources, economic basis and traffic conditions and other geographical advantages, have developed into a relatively high level of tourism in Hainan Island as a demonstration destination. The tourism development efficiency of Tunchang county and Ledong County is below 0.6, which is seriously behind the average of Hainan Island. The tourism industry in Baoting county and Qiongzhong County started late, the scale of tourism industry is limited, and the environmental health pressure brought by tourism development is small. Lingao County and Tunchang County have a small ecological environment cost per unit tourism output value, which is one of the cities with a better relationship between tourism development and ecological environment protection.

The improvement in tourism development efficiency in Hainan Island is affected by many factors, among which knowledge innovation is the main obstacle. Although Hainan has a long history of tourism development, its tourism form is relatively simple; the tourism brand effect is limited; and it has not formed a stable growth of consumer groups in the international tourism market. In particular, the application technology level of the tourism industry, the management mode of the tourism industry, and the construction and introduction of tourism talents are still insufficient, and it is difficult to compete with Bali, Phuket island, Hawaii, and other destinations with similar climate conditions. Urban fixed asset investment, tourism resource endowment, and knowledge innovation are the main factors to improve the efficiency of local tourism development. However, the spatial spillover effect of Hainan’s overall tourism development is not obvious, and some factors even inhibit the development of tourism in neighboring areas. This is mainly caused by the imperfect connection network between cities, the homogenous competition of tourism products, and the unbalanced flow of tourism elements. In the geographic distance weighting matrix and economic distance weighting matrix measure space overflow, due to the differences in emphasis, between in addition to tourism resource endowment has more significant spatial spillover effect, other indicators of direction of the overflow effect works even have differences; this also confirmed that the tourism resources endowment is the basis of regional tourism. To improve the tourism development efficiency of Hainan Island, it is necessary to target Bali Island, Phuket island, and other internationally famous tourist destinations, which reasonably increase the input of tourism factors, actively introduce advanced management mode and technical personnel, and constantly improve the international influence and market competitiveness of “Hainan International Tourism Island.”

### 6.2. Limitations and Future Research

This study has the following two limitations. First, based on the construction of tourism development efficiency of input and output index system, while considering the effects of knowledge innovation and the ecological environment pollution output, but considering city traffic conditions and policy differences, it failed to fully be included in the input index system within the scope of the research result to a certain extent, resulting in objective deviation to some extent. Second, this study took Hainan Island of China as the case site. Although it is typical to some extent, it has limited reference value for other tourist destinations in the world and cannot form universal research results and construction significance. Future research should be long-term tracking a typical case to tourism development, continuous correction efficiency, and space overflow model tools; should improve the evaluation index system of tourism destination development efficiency; focus on tourist destinations’ worldwide comparative study; make it from the specific situation and geographical restrictions; refine the general rule of world tourism destination development efficiency; and promote the sustainable development of world tourism.

## Figures and Tables

**Figure 1 ijerph-19-03755-f001:**
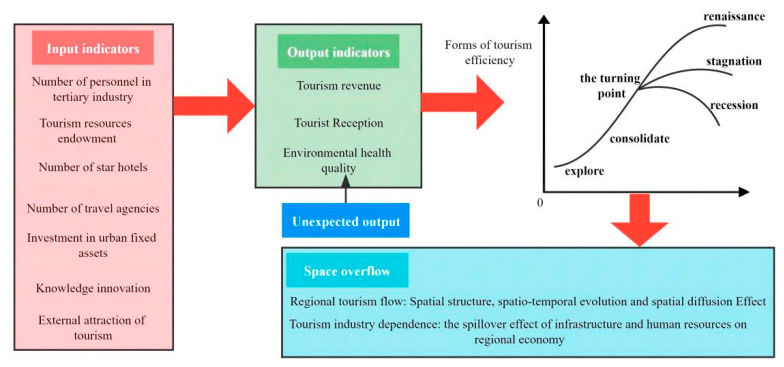
Tourism development efficiency and spatial spillover performance chart.

**Figure 2 ijerph-19-03755-f002:**
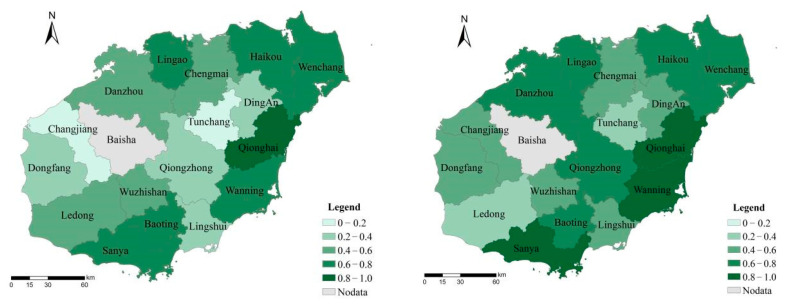
The evolution of the spatial and temporal pattern of tourism development efficiency of Hainan Island. In 2001, 2005, 2010, 2015 and 2020.

**Figure 3 ijerph-19-03755-f003:**
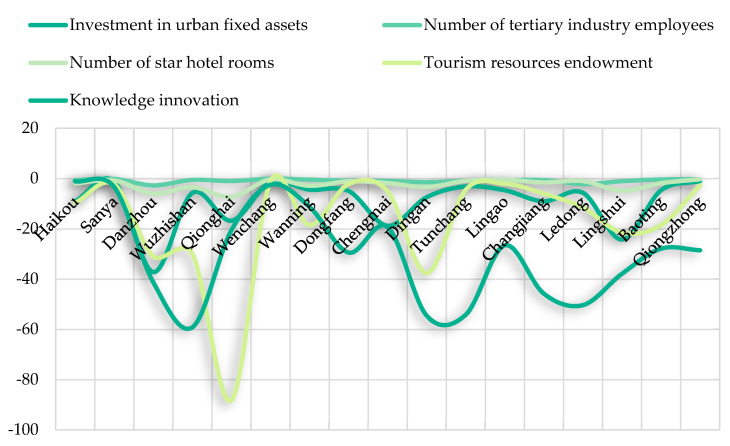
Potential indicators of tourism development efficiency improvement in Hainan cities from 2001–2020.

**Table 1 ijerph-19-03755-t001:** Input–output indexes of tourism development efficiency in Hainan Island.

The Index Type	Index	Variable	Unit
Input indicators	Number of tertiary industry employees	X_1_	Ten thousand people
	Investment in urban fixed assets	X_2_	Ten thousand yuan
	Number of star hotel rooms	X_3_	The room
	Tourism resources endowment	X_4_	-
	Knowledge innovation	X_5_	-
Output indicators	Tourist Reception	Y_1_	Person-time
	Total income from tourism	Y_2_	One hundred million yuan
	Environmental health quality	Y_3_	%

**Table 2 ijerph-19-03755-t002:** Efficiency of tourism development in Hainan Island under different models from 2001 to 2020.

Period	TE:EBM-I-C	PTE:EBM-I-V	SE:EBM-I-V	CCR-I-C	SBM-I-C
2001	0.6561	0.9671	0.6697	0.7243	0.5458
2002	0.6367	0.9636	0.6532	0.7314	0.5060
2003	0.4964	0.9513	0.5131	0.5514	0.4170
2004	0.7183	0.9626	0.7368	0.8125	0.5977
2005	0.6963	0.9607	0.7169	0.8081	0.5596
2006	0.7093	0.9440	0.7458	0.8219	0.5578
2007	0.7058	0.9667	0.7305	0.8679	0.5132
2008	0.7684	0.9668	0.7941	0.9110	0.6110
2009	0.7172	0.8886	0.8054	0.8362	0.5398
2010	0.6886	0.9804	0.6980	0.7488	0.6098
2011	0.5302	0.9486	0.5551	0.5869	0.4407
2012	0.8699	0.9514	0.9169	0.9142	0.7975
2013	0.8538	0.9228	0.9242	0.8957	0.7861
2014	0.8451	0.9216	0.9150	0.8777	0.7982
2015	0.9246	0.9614	0.9580	0.9511	0.8875
2016	0.8148	0.9214	0.8753	0.8536	0.7524
2017	0.7525	0.9045	0.8318	0.7893	0.6820
2018	0.9149	0.9440	0.9662	0.9421	0.8593
2019	0.8708	0.9481	0.9173	0.9091	0.8057
2020	0.7010	0.8366	0.8392	0.7509	0.5733
Mean value	0.7435	0.9406	0.7881	0.8142	0.6420

**Table 3 ijerph-19-03755-t003:** Tourism development efficiency, pure technical efficiency, and scale efficiency in Hainan Island from 2001 to 2020.

City	TE:EBM-I-C	PTE:EBM-I-V	SE:EBM-I-V
Haikou	0.9782	0.9795	0.9984
Sanya	0.9942	1.0000	0.9942
Danzhou	0.6596	0.8321	0.8073
Wuzhishan	0.6035	0.9543	0.6348
Qionghai	0.7788	0.8696	0.8983
Wenchang	0.9510	0.9666	0.9840
Wanning	0.9444	0.9658	0.9695
Dongfang	0.6808	0.9483	0.7154
Chengmai	0.8361	0.9148	0.9044
Dingan	0.5591	0.8729	0.6361
Tunchang	0.5712	0.9781	0.5832
Lingao	0.7846	0.9504	0.8177
Changjiang	0.6193	0.9399	0.6464
Ledong	0.5853	0.9136	0.6528
Lingshui	0.6685	0.9233	0.7177
Baoting	0.6946	0.9847	0.7053
Qiongzhong	0.7310	0.9965	0.7327
Total	0.7435	0.9406	0.7881

**Table 4 ijerph-19-03755-t004:** Estimation results of spatial panel Dubin model.

Variable	Geographical Distance Weight Matrix		Economic Distance Weight Matrix	
	Coefficient	*p* Values	Coefficient	*p* Values
lnIIUFA	0.5098 **	0.037	0.4091	0.196
	(2.09)		(1.29)	
lnNOTIE	1.0871 ***	0.000	0.3426	0.257
	(3.86)		(1.13)	
lnNOSHR	0.2354	0.141	−0.0817	0.655
	(1.47)		(−0.45)	
lnTRE	1.1202 ***	0.000	0.9643 ***	0.000
	(8.74)		(6.49)	
lnKI	0.8646	0.167	1.4382 *	0.072
	(1.38)		(1.80)	
W*lnIIUFA	−2.6000	0.119	3.8044 ***	0.000
	(−1.56)		(5.38)	
W*lnNOTIE	10.3377 ***	0.000	− 0.6950	0.274
	(5.39)		(−1.09)	
W*lnNOSHR	0.3480	0.724	−1.6883 ***	0.000
	(0.35)		(−3.57)	
W*lnTRE	5.0744 ***	0.000	0.8831 **	0.015
	(5.99)		(2.43)	
W*lnKI	−4.388	0.225	5.3116 **	0.014
	(−1.21)		(2.44)	
Space effect	control		control	
Time effect	control		control	
Log-likelihood	827.6163		855.2701	
R party	0.4356		0.4537	

Note: T value in parentheses; ***, **, * represent 0.01, 0.05, 0.1 significance level respectively.

**Table 5 ijerph-19-03755-t005:** Partial differential estimation results of spatial spillover effect.

	Variable	Direct Effect		Indirect Effect		Total Effect	
		Coefficient	*p* Values	Coefficient	*p* Values	Coefficient	*p* Values
Geographical distance weight matrix	lnIIUFA	0.8603 ***	0.002	−1.4939 **	0.023	−0.6335	0.283
		(3.14)		(−2.28)		(−1.07)	
	lnNOTIE	0.3044	0.321	3.3518 ***	0.000	3.6563 ***	0.000
		(0.99)		(4.43)		(5.28)	
	lnNOSHR	0.2668	0.170	−0.0814	0.834	0.1854	0.555
		(1.37)		(−0.21)		(0.59)	
	lnTRE	0.8425 ***	0.000	1.1754 ***	0.001	2.0179 ***	0.000
		(5.73)		(3.32)		(6.92)	
	lnKI	1.4576 **	0.037	−2.5965 *	0.072	−1.1389	0.366
		(2.08)		(−1.8)		(−0.9)	
Economic distance weight matrix	lnIIUFA	0.2450	0.468	3.2316 ***	0.000	3.4766 ***	0.000
		(0.73)		(5.5)		(6.16)	
	lnNOTIE	0.3652	0.225	−0.6770	0.236	−0.3118	0.628
		(1.21)		(−1.18)		(−0.48)	
	lnNOSHR	0.0143	0.936	−1.4727 ***	0.000	−1.4585 ***	0.000
		(0.08)		(−3.86)		(− 3.63)	
	lnTRE	0.9201 ***	0.000	0.6116 **	0.034	1.5317 ***	0.000
		(6.68)		(2.13)		(5.02)	
	lnKI	1.2244	0.109	4.4281 **	0.016	5.6525 ***	0.01
		(1.6)		(2.4)		(2.56)	

Note: T values are in parentheses, and all are spatio-temporal double fixed results; ***, **, * represent 0.01, 0.05, and 0.1 significance levels, respectively.

## Data Availability

The datasets used and/or analyzed during the current study are available from the corresponding author on reasonable request.

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
