# Peer review of "Evaluation of Tourism Development Efficiency and Spatial Spillover Effect Based on EBM Model: The Case of Hainan Island, China"

_ijerph, 2022, doi:10.3390/ijerph19073755_

Round 1

Reviewer 1 Report

Dear authors, your work is devoted to a very interesting and significant topic today - the development of tourist destination. Your research is original. But your findings may be more relevant to science if you try to clarify a number of points:

1. How did the pandemic period affect the development of tourism in your destination

2. What impact the Chine goverment policy has on the development of the tourism sector in Taiwan

3. what direction of development of your tourism destination (strategies of development of tourism) needs to be chosen

I also want to note that the presence of a significant amount of digital material in the article is not always good, since attention is distracted from the text.

Author Response

I would like to thank all of the experts for their professional review of
Evaluation of tourism development efficiency and spatial spillover effect
based on EBM model: The case of Hainan Island, China (ijerph-1624012), and
thank the editorial department for their support and help in this
manuscript.This article has been revised according to the expert review
opinions,

Reviewer 2 Report

I admit that I have mixed feelings about this article because it is undoubtedly a mature scientific work and deals with an interesting topic. My doubts are raised by references to previous research on this subject - this is especially visible in the Discussion chapter, which is extremely modest compared to other chapters. The authors did not discuss the most popular model of development of tourist areas by R.W. Buter, although there is a graphic reference to this method (Fig. 1).
I believe that the article should be corrected in terms of reviewing commonly known methods of describing the development of tourist areas, including Tourism Area Life Cycle (TALC).
In the discussion, it is worth comparing the ability of the developed method and other methods known in the literature.

Author Response

I would like to thank all of the experts for their professional review of Evaluation of tourism development efficiency and spatial spillover effect based on EBM model: The case of Hainan Island, China (ijerph-1624012), and thank the editorial department for their support and help in this manuscript.This article has been revised according to the expert review opinions, Please see the attachment.

Thank you again for your professional advice, which greatly improved the quality of the article!I am also grateful to the expert teachers and the editorial department teachers for the opportunity to modify this article!

Best wishes!

Yours sincerely

Pengfei Zhang, on behalf of the co-authors

18 Mar 2022
